# PLAN-RAG: PLANNING-GUIDED RETRIEVAL AUGMENTED GENERATION

## ABSTRACT

We introduce Planning-guided Retrieval Augmented Generation (Plan-RAG), a novel framework that augments the *retrieve-then-reason* paradigm of existing RAG frameworks to *plan-then-retrieve*. Plan-RAG formulates a reasoning plan as a directed acyclic graph (DAG), decomposing queries into interrelated atomic sub-queries. Answer generation follows the DAG structure, allowing significant gains in efficiency through parallelized retrieval and generation. While state-of-the-art RAG solutions require extensive data generation and fine-tuning of language models (LMs), Plan-RAG incorporates frozen LMs as plug-and-play experts to generate high-quality answers. Compared to existing RAG solutions, Plan-RAG demonstrates significant improvements in reducing hallucinations and bolstering attribution due to its structured sub-query decomposition. Overall, Plan-RAG offers a new perspective on integrating external knowledge in LMs while ensuring attribution by design, contributing towards more reliable LM-based systems.

## 1 INTRODUCTION

Despite the remarkable success of Large Language Models (LLMs) across various domains (Torfi et al., 2020; Zhao et al., 2023; Brown et al., 2020), LLMs face critical challenges that impede their widespread adoption in critical applications such as healthcare and finance (Pal et al., 2023; Zhao et al., 2024). Among these challenges, hallucination stands out as a particularly pressing concern (Ji et al., 2023; Maynez et al., 2020). Hallucination in LLMs, as defined by Rawte et al. (2023), occurs when model generation deviates from factual information or includes false statements, potentially leading to misinformation and compromised decision-making.

Retrieval Augmented Generation (RAG, Petroni et al., 2020; Lewis et al., 2020; Guu et al., 2020) has emerged as a promising framework to address hallucinations in LLMs by integrating external information. RAG aims to ground the generation in factual information, theoretically reducing the likelihood of generating false content. The standard RAG framework follows the *retrieve-then-reason* paradigm: first, documents are retrieved based on an input query, and then the LLM reasons over them to generate a response. However, a recent study by Shuster et al. (2021)

**Query:**
```
In what year was the coach who led
the 2007 South Carolina Gamecocks
football team in his third season as
USC head coach born?
```

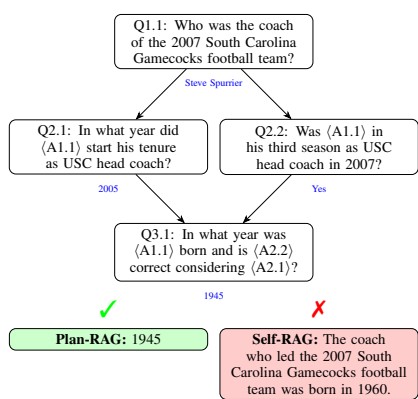

Figure 1: **HotpotQA example:** Plan-RAG's reasoning DAG and final generation compared to Self-RAG's output, illustrating the benefits of *plan-then-retrieve* approach.

revealed that RAG systems are not immune to hallucination, particularly when the retrieved documents are irrelevant, relevant but insufficient, or exceed the context window, preventing the LLM from effectively reasoning over all of them. Another critical limitation that closely relates to hallucination is lack of attribution (Rashkin et al., 2023; Bohnet et al., 2022). While RAGs can access external information, they often struggle to reliably link their generated content to specific retrieved documents, undermining the system's trustworthiness and interpretability (Xia et al., 2024; Qi et al., 2024). This interplay between hallucination and lack of attribution presents a significant challenge in developing reliable LLM systems (Ji et al., 2023; Adewumi et al., 2024; Huang et al., 2023).

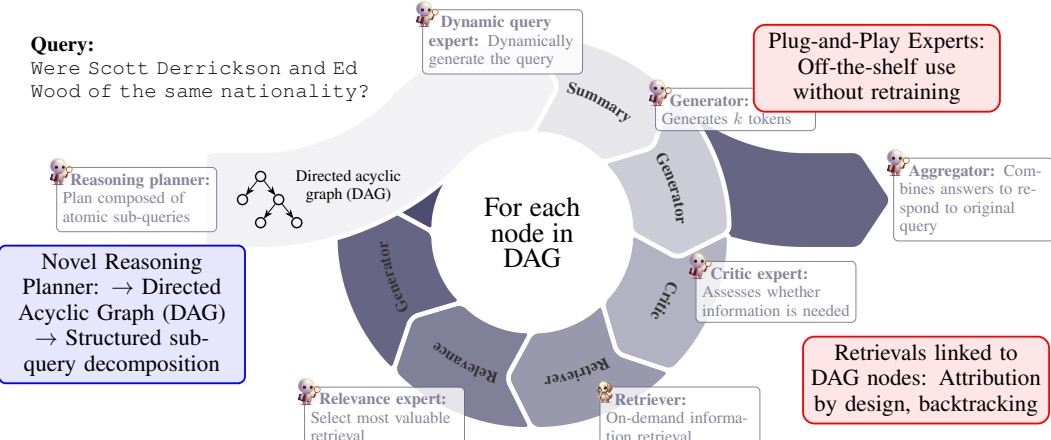

Figure 2: **Plan-RAG:** Plan-RAG's core novelty lies in the generation of a reasoning Directed Acyclic Graph (DAG) plan that decompose queries into structured sub-queries, enabling efficient, parallelized retrieval and generation. Additionally, *plug-and-play* experts ensure high-quality retrieval and consistency, reducing hallucinations and ensuring attribution by design.

To overcome these challenges, we propose a novel framework—Planning-guided Retrieval Augmented Generation (Plan-RAG). We augment the conventional *retrieve-then-reason* paradigm to *plan-then-retrieve* paradigm, fundamentally altering how LLMs interact with external knowledge. Unlike traditional query decomposition methods such as RA-ISF (Liu et al., 2024) and RQ-RAG (Chan et al., 2024) that generate isolated or sequential sub-queries. Plan-RAG formulates a comprehensive reasoning plan represented as a directed acyclic graph (DAG). This reasoning DAG decomposes the main query into interrelated *atomic* sub-queries, providing a computational structure that enables efficient information sharing between sub-queries (see Fig. 1). In addition, the reasoning DAG characterizes conditional independence relationships between queries, facilitating parallelization, resulting in more efficient generation. Moreover, it provides improved explainability and debuggability, attribution, and efficient backtracking for correcting specific segments of the responses. We cover these in detail in Sec. 3.1 and showcase the comparison of the key features between recently proposed RAG frameworks and Plan-RAG in Table 1.

In addition, unlike recent proposals for improving RAG (Asai et al., 2023; Chan et al., 2024; Liu et al., 2024), Plan-RAG avoids the expensive step of finetuning an LM. It can work with any pretrained LM, by incorporating a set of independent *plug-and-play* experts: critic expert, relevance expert, *etc.*. The critic expert enables on-demand retrievals, assessing when additional information is needed. The relevance expert refines the retrieval process, ensuring the most relevant documents are selected. Notably, the critic expert enables on-demand retrievals, while the atomic nature of the sub-queries limits retrieval to a single document. These features inherently promote attribution and reduce hallucination. In Sec. 3.2, we discuss the experts and their benefits in detail.

**Contributions** *(i)* We propose augmenting the *retrieve-then-reason* paradigm to a *plan-then-retrieve* paradigm, offering a new perspective on integrating external knowledge in LLMs. *(ii)* We introduce a reasoning DAG that inherently enhances attribution and debuggability capability. *(iii)* We demonstrate reduced hallucinations attributed to the *plug-and-play* experts and atomic nature of sub-queries. *(iv)* We present a general and practical framework, as Plan-RAG uses a frozen LLM without any assumptions.

## 2 RELATED WORK

Retrieval-augmented generation (RAG) enhances large language models (LLMs) by integrating relevant external documents, leading to notable performance improvements, particularly in knowledge-intensive tasks (Lewis et al., 2020; Guu et al., 2020). Retrieval strategies in RAG models can be categorized into three paradigms based on the frequency of retrievals: (1) one-time retrieval, (2) retrieval every $k$ tokens, and (3) adaptive retrieval. Models employing one-time retrieval include DrQA (Chen et al., 2017), REALM (Guu et al., 2020), and ATLAS (Izacard et al., 2023). Retrieval at fixed

Table 1: **Comparison of key features across various RAG frameworks:** The proposed Plan-RAG framework demonstrates advantages in reliable information flow, parallelization, debuggability, attribution, and ease of use without finetuning, compared to two existing advanced RAG frameworks.

|  | **Vanilla RAG** | **Self-RAG** | **RQ-RAG** | **Plan-RAG (Ours)** |
|---|---|---|---|---|
| Relevant flow of information | ✗ | ✓ | ✗ | ✓ |
| Parallelization | ✗ | ✗ | ✗ | ✓ |
| Debuggability & backtracking | ✗ | ✗ | ✗ | ✓ |
| Attribution | ✗ | ✗ | ✗ | ✓ |
| Off-the-shelf use (no finetuning) | ✓ | ✗ | ✗ | ✓ |

intervals (every $k$ tokens) is used by RALM (Ram et al., 2023), RETRO (Borgeaud et al., 2022), and InstructRetro (Wang et al., 2024a). In contrast, adaptive retrieval approaches—such as Self-RAG (Asai et al., 2023), SPALM (Yogatama et al., 2021), Adaptive kNN (Drozdov et al., 2022), and Active-Retriever (Jiang et al., 2023)—dynamically adjust the frequency and nature of document retrieval based on task requirements and input context. FLARE (Jiang et al., 2023) is a framework that uses token probability distributions to trigger retrievals and predict temporary next sentences, enhancing the quality of subsequent retrievals. SPALM (Yogatama et al., 2021) incorporates additional trained components to manage adaptive retrieval at the token level. RETRO (Borgeaud et al., 2022) performs document retrieval at every $k$ tokens, necessitating the training of a specialized architecture. Similarly, kNN-LM (Khandelwal et al., 2019) uses k-nearest neighbor searches on embeddings to calculate token probabilities, with are then aggregated with model outputs, adding latency.

Recently, Wang et al. (2024b) proposed RAFT, a framework designed to address the challenges of retrieval and hallucination by combining chain-of-thought (CoT) reasoning with RAG. It initializes a set of preliminary thoughts and iteratively loops over them, performing retrievals based on the current thought and generation to incrementally refine the output. Asai et al. (2023) introduced Self-RAG, a framework in which an LLM is trained on an extended vocabulary set for retrieval and evaluation. These new tokens are generated using the GPT-4 model, and a critic LLM is then trained on this supervised dataset to enhance performance. Liu et al. (2024) propose RA-ISF, an architecture that combines multiple specialized models that are trained on specific datasets. Chan et al. (2024) proposed RQ-RAG, where an LLM is equipped with capabilities for query rewriting, decomposition, and disambiguation. This helps in handling ambiguous or complex queries more effectively resulting in improved performance. In contrast to these approaches, which often necessitate fine-tuning or specialized training, Plan-RAG proposes a formal reasoning DAG and a set of off-the-shelf experts, offering a flexible solution without the need for finetuning.

Mishra et al. (2024) propose an editor model that processes the generator output and corrects hallucinations by incorporating factual information based on retrieved data. The editor is an SLM compared to the generator and is trained on a custom dataset. Similarly, Gou et al. (2023) introduce the CRITIC framework, which interacts with external web tools to refine LLM outputs and minimize hallucinations. Asai et al. (2022) suggest improving retriever accuracy by embedding task-specific instructions within the query, rather than employing different retrievers for various data types (*e.g.*, code, questions). Recently, Dalal & Misra (2024) explore using the entropy of the output token distribution to detect when a model is likely hallucinating. Extending these approaches, we propose Plan-RAG, which addresses the performance, hallucination, and attribution issues in the RAG framework.

## 3 PLAN-RAG

Consider the query from Fig. 3: *"What is the distance between the locations that hosted the last two Men's Cricket World Cup finals?"*. Conventional RAG frameworks struggle with such complex, multi-hop queries as they retrieve information only once at the start of the generation process, following a *retrieve-then-reason* approach. Although recently proposed query decomposition methods (Chan et al., 2024; Liu et al., 2024) aim to address this issue, they rely on simple structures such as sequential or independent queries. As a result, they fail to capture the inherent reasoning structure.

To overcome these limitations, we propose Planning-guided Retrieval-Augmented Generation (Plan-RAG), which adopts a *plan-then-retrieve* paradigm. Given a query, Plan-RAG generates a reasoning plan upfront, as illustrated in Fig. 3, decomposing the query into *atomic* subqueries. In the reasoning

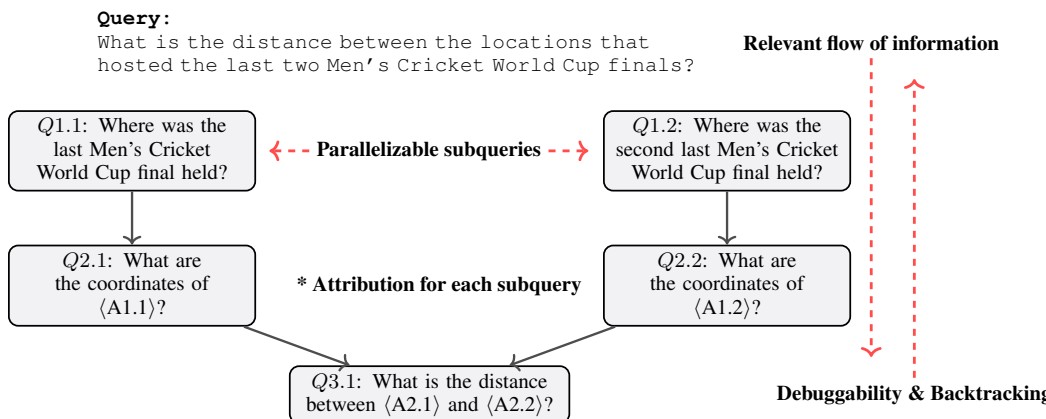

Figure 3: **Reasoning plan example:** A Reasoning DAG generated by the reasoning plan expert, highlighting key advantages: only relevant information flows to each subquery, subqueries on the same depth can be executed in parallel, attribution is inherent by design, and the DAG structure allows for debugging and backtracking.

DAG presented, subqueries *Q1.1* and *Q2.1* can be processed independently of *Q1.2* and *Q2.2*, while *Q3.1* depends only on the answers *A2.1* and *A2.2*. Query decomposition methods fail to exploit such structure. The reasoning plan is structured using a Directed Acyclic Graph (DAG), allowing for parallelizable subqueries, providing a way for relevant flow of information, and backtracking. The term *atomic* refers to subqueries that request a single piece of information and thus can be answered by a single document—breaking them further will not help. By ensuring a single retrieved document per subquery, Plan-RAG provides attribution *by design*, as the generated text for each subquery can be solely attributed to the singleton retrieved document.

Once the DAG is created, the generator LM processes each graph node in topological order. Plan-RAG introduces a set of *plug-and-play* experts to control generation at each node (see Fig. 2). These experts generate dynamic subqueries, access the need for retrieval, and identify relevant document(s) for each subquery. The experts are invoked after every $k$-tokens generated by the generator LM or when a stop token (such as end of sentence) is reached. Once an answer is generated for a subquery node, the processing moves to its child nodes, continuing until all leaf nodes are reached. A key advantage of Plan-RAG is modularity: it is compatible with any generator LM and any implementation of the experts. The full algorithm for Plan-RAG is described in Alg. 1 and Alg. 2.

## 3.1 REASONING PLAN: DAG

At the core of Plan-RAG is a reasoning plan represented as a Directed Acyclic Graph (DAG), generated by an LLM using a suitable prompt (see App. C.2). The DAG is structured so that the root nodes can be answered independently, while each subsequent node can be answered based on the answers to its parent nodes. Thus, the reasoning plan follows Markov assumption. Formally, the reasoning DAG is represented as $\mathcal{G}(\mathbf{V}, \mathbf{E})$ where $\mathbf{V}$ represents the set of generated subqueries for a given query, and $\mathbf{E}$ denotes the edges. The answer to any subquery $q \in \mathbf{V}$ can be computed as,

$$G(q) = f(G(\mathbf{Pa}(v)), q, \mathbf{D}_q); \quad \mathcal{G} = f_{\text{Reasoning}}(Q) \tag{1}$$

where $\mathbf{Pa}(q)$ refers to all parents of $q$ in the DAG and $\mathbf{D}_q$ represents the retrieved documents for $q$, if any, and $Q$ is the main query. The function $f$ (typically the generator LM, $\mathbf{f}_{\text{Generator}}$) generates the response text using the subquery, the responses from its parent subqueries, and the document retrievals. When applied recursively, $f$ provides a mechanism to generate the response to a query by traversing the DAG from the root nodes to the leaves. In practice, $G(v)$ may be computed in an auto-regressive manner with potentially multiple retrievals after each $k$-tokens as described above.

An example of the reasoning DAG is shown in Fig. 3. Each node is numbered as $\langle i.j \rangle$ where $i$ refers to the depth of the node from the root and $j$ refers to its index among the nodes at the same depth. A notable feature of the reasoning DAG is the use of a special tag $\langle \mathbf{AI.J} \rangle$ that enables dynamic subquery generation. In the tag $\langle \mathbf{AI.J} \rangle$, $\mathbf{I}$ and $\mathbf{J}$ are the integer values representing the Question IDs that are

required to complete the subquery. For example, in Fig. 3, subquery *Q2.1* depends on the answer to the subquery *Q1.1*; the special tag $\langle$A1.1$\rangle$ allows filling in of the answer dynamically at run time.

### 3.2 EXPERTS

Plan-RAG incorporates a set of independent, *plug-and-play* experts that enhance its capabilities and helps in addressing key challenges in traditional RAG systems. These experts work in concert to boost the accuracy, reliability, relevance, and interpretability of LLM-generated responses. Below, we detail the role of each expert and how they contribute to the overall framework's performance.

1. **Dynamic Query Expert:** The dynamic query expert is responsible for generating subqueries by embedding the answers into the corresponding tags within the subquery. In addition to generating the subquery, it captures the Markovian dependencies between subqueries, ensuring that only the relevant information is passed while irrelevant details are abstracted. Formally, the process is defined as $\tilde{q} \leftarrow \mathbf{E}_{\text{DynamicQuery}}(q, \mathbf{T_q})$, where $\tilde{q}$ is the dynamically generated subquery, $q$ is the subquery within the DAG containing the special tag $\langle$**AI.J**$\rangle$, and $\mathbf{T_q}$ represents the set of question-answer pairs associated to the special tags in $q$.

2. **Critic Expert:** The critic expert enables on-demand retrievals by assessing when additional information is needed during the generation process. This expert plays a crucial role in identifying knowledge gaps, triggering retrievals, and reducing hallucination. By analyzing the current context and the generation task, the critic determines when the LLM lacks sufficient information to provide an accurate response. Upon identification, the critic initiates the retrieval process, ensuring that the system acquires the necessary information dynamically. Formally, $C \leftarrow \mathbf{E}_{\text{Critic}}(G, \tilde{q})$, where $G$ is the current generation, $q$ is the (sub)query, and $C$ is a boolean representing whether retrieval is required or not. The critic expert can be configured to run after one sentence or $k$-tokens.

3. **Relevance Expert:** The relevance expert refines the retrievals, ensuring the selection of the most relevant document to the subquery at hand. Its key tasks include relevance scoring, ranking and selection. The expert reduces the context window usage by ranking and selecting only the most relevant documents. Formally, $\mathbf{r}^\star \leftarrow \mathbf{E}_{\text{Relevance}}(G, \tilde{q}, \mathbf{r})$, where $G$ is the current generation, $\tilde{q}$ is the (sub)query, $\mathbf{r}$ is the set of retrieved document, and $\mathbf{r}^\star$ is the set of relevant documents obtained from the retriever $\mathbf{f}_{\text{Retriever}}$. The relevance can be configured to output either a single document or a set of relevant documents, depending on the task requirements.

4. **Aggregator:** The aggregator expert combines multiple answers to a set of subqueries to generate a cohesive and comprehensive response to the original query. It plays a key role in subquery integration and ensuring a balanced, holistic final response. The expert ensures that the final response addresses all aspects of the original query in a balanced and thorough manner. Formally, $G \leftarrow \mathbf{E}_{\text{Aggregator}}(\mathbf{q}, \mathbf{G})$, where $G$ is the combined generation, $\mathbf{q}$ is the set of queries, and $\mathbf{G}$ is the set of their respective generations.

All these experts work in tandem to create a robust and adaptive system, where each component plays a clear role in improving the overall performance of Plan-RAG. Plan-RAG reduces hallucination by incorporating on-demand retrievals and relevance expert, decreasing the likelihood of generating false or unsupported information. The system improves attribution by more easily tracing generated content back to specific retrievals, enhancing the interpretability and trustworthiness of the outputs. Plan-RAG ensures enhanced relevance through expert-guided retrieval and relevance expert, guaranteeing that the most pertinent information is used in the generation process.

As shown in Alg. 1 and Alg. 2, in the default setting, Plan-RAG involves 1 GPT-4o class to get the *reasoning DAG* and then for each node, the *critic expert* is called after every $k$-tokens (or at end-of-sentence). If it predicts that retrievals are needed, then the *retriever* is invoked and subsequently the *relevance expert* is called. The *dynamic query expert* is only called once per subquery in case it contains a special dependency tag, and the *aggregator expert* is called once for the entire query. Therefore, for a reasoning DAG with n nodes, we expect $O(2ns + n)$ calls where $s$ is the number of sentences/$k$-length phrases in the generation per node.

### 3.3 BENEFITS OF PLAN-RAG

The breaking up of a query into a reasoning DAG approach offers several advantages over existing RAG methods (see Table 1). We elaborate on these advantages in detail below.

**Algorithm 1** Plan-RAG framework

*Input:* $Q, k$: Query, generation-token size
*Output:* Generation $G$
Get a reasoning plan: $\mathbf{q} \leftarrow f_{\text{Reasoning}}(Q)$
Identify root nodes: $\mathbf{q}_{root}$
Calculate depth of each node from root:
$l_q \leftarrow maxdist(q, \mathbf{q}_{root})$
**for** i: 0 to $\max_q(l_q)$ **do**
    **for** parallel: $q$ in $\{q : l_q = i\}$ **do**
        Get parent questions and answers:
        $M_q \leftarrow \mathbf{Pa}(q) \ \& \ M_a \leftarrow G_{M_q}$
        Dynamically generate the subquery:
        $\tilde{q} \leftarrow \mathbf{E}_{\text{DynamicQuery}}(q, M_q, M_a)$
        Obtain generated answer for query $q$
        by calling Alg. 2 with inputs $\tilde{q}, k$.
    **end for**
**end for**
$G \leftarrow \mathbf{E}_{\text{Aggregator}}(\mathbf{q}, G_1, G_2, \ldots, G_q)$
**return** $G$

**Algorithm 2** Generate answer

*Input:* $\tilde{q}, k$
Initialize $G_q$ with empty string and $\mathbf{r}^\star \leftarrow \phi$
**while** generation not finished **do**
    Generate $k$ tokens: $\bar{G}_q \leftarrow \mathbf{f}_{\text{Generator}}(G_q, \tilde{q}, \mathbf{r}^\star)$
    Check if retrieval is required:
    $C \leftarrow \mathbf{E}_{\text{Critic}}(\bar{G}_q, \tilde{q}, \mathbf{r}^\star)$
    **if** C **then**
        Get retrievals: $\mathbf{r} \leftarrow \mathbf{f}_{\text{Retriever}}(\bar{G}_q, \tilde{q})$
        **if** retrieval $\mathbf{r}$ is non trivial **then**
            Get the relevant retrieval(s):
            $\mathbf{r}^\star \leftarrow \mathbf{E}_{\text{Relevance}}(\bar{G}_q, \tilde{q}, \mathbf{r})$
        **else**
            Re-write the query & retry retrievals.
        **end if**
    **end if**
    $G_q \leftarrow G_q \bigoplus \bar{G}_q$
**end while**
**return** $G_q$

**Attribution by design:** The atomic nature of sub-queries improves attribution by limiting retrieval to a single document per generation. In its default configuration, the relevance is constrained to select only one relevant document per subquery, defined as $\mathbf{r}^\star = \mathbf{E}_{\text{Relevance}}(\bar{G}_q, \tilde{q}, \mathbf{r})$, where $|\mathbf{r}^\star| = 1$. This ensures that each subquery generation is directly linked to a single retrieved document, establishing a clear, one-to-one mapping between the document and the subquery's response. This setup guarantees attribution by design, allowing easy traceability of each generated response back to its specific source document. When the relevance expert is permitted to retrieve multiple documents, this direct attribution feature diminishes. However, empirical results indicate that even in these configurations, the relevance expert typically selects only one document. This is largely due to the atomic nature of the subqueries, where the relevant information for each subquery tends to be contained within a single document. We showcase this with an experiment which we discuss in detail in App. E.1.

**Efficiency:** The reasoning DAG enhances efficiency by leveraging queries that are on the same depth within the DAG, or on the independent paths in the DAG. This significantly reduces latency, and improves context handling through relevant flow of information *i.e.* abstracting unnecessary information such as subqueries, their responses, retrieved data that are not related. Plan-RAG achieves greater **efficiency in context utilization** as compared to vanilla RAG models as well as RAG frameworks like RA-ISF and RQ-RAG. By employing a focused approach where typically only one highly relevant retrieval is used, Plan-RAG maximizes the use of the limited context window available to SLMs. This targeted use of context allows the system to handle more complex queries and maintain coherence over longer interactions without the need for extensive context management.

**Debuggability:** The reasoning DAG provides a robust mechanism for identifying and rectifying erroneous generations by backtracking through the paths from the leaf nodes to the root node. This allows us to analyze the subqueries and their corresponding responses at each step. Upon identifying the error node, we can address the issue by providing additional context or clarifying the subquery, among other strategies. After adjustments, we regenerate the outputs only for the affected path and rerun the aggregator. This iterative debugging process enhances the explainability of the RAG system.More broadly, the reasoning DAG maps how various pieces of information contribute to the final generation.We demonstrate this *debug-and-backtrack* capability in Fig. 4, where Plan-RAG initially generates an incorrect response. However, by analyzing the DAG and its corresponding subqueries, we are able to identify and rectify the error, ultimately leading to a correct generation.

## 4 EXPERIMENTS

We conduct a comprehensive series of experiments to demonstrate the capabilities and efficiency of the proposed framework, Plan-RAG. Our experiments highlight Plan-RAG's improved accuracy, attribution, and debuggability. Additionally, we perform ablation studies to analyze the contribution of each expert module and quantify their individual impact on the overall performance of Plan-RAG.

Table 2: **Reasoning DAG depth** (percentage/count) for multi-hop query (HotpotQA, StrategyQA, Arc-Processed) and single-hop query (PopQA) data sets. For single-hop queries, the DAG primarily has a depth of 0, which is desirable, while multi-hop queries typically require deeper reasoning paths.

| Dataset | Depth 0 | Depth 1 | Depth 2 | Depth 3 | Depth$\geq$4 |
|---|---|---|---|---|---|
| **HotpotQA** (Multi-hop) | 0.5% (35) | 12.8% (939) | 79.5% (5848) | 6.8% (501) | 0.4% (29) |
| **StrategyQA** (Multi-hop) | 0.9% (22) | 42.9% (960) | 51.2% (1145) | 4.5% (101) | 0.3% (6) |
| **Arc-Processed** (Multi-hop) | 0.1% (2) | 72.1% (790) | 25.5% (279) | 2.1% (23) | 0.0% (1) |
| **PopQA** (Single-hop) | 77.9% (1090) | 0.8% (11) | 18.9% (264) | 2.4% (34) | 0.0% (0) |

**Datasets** We evaluate Plan-RAG on four datasets: HotpotQA (Yang et al., 2018), StrategyQA (Geva et al., 2021), and ARC-challenge (Clark et al., 2018) consisting of multi-hop queries; and PopQA (Mallen et al., 2022) consisting of single-hop queries. The dataset details are given in App. A.

**Baselines and competing methods** We evaluate the performance of Plan-RAG method against two recently proposed RAG methods that have achieved state-of-the-art results on the datasets above: Self-RAG (Asai et al., 2023) and RQ-RAG (Chan et al., 2024), outperforming SAIL (Luo et al.), Toolformer (Schick et al., 2024), Alpaca models, and proprietary LLMs like Perplexity.ai and Chat-GPT. We use the officially released code and associated models for both of these methods. Details are present in App. C.1. We also compare with the following baseline methods: (1) Vanilla LLMs (*i.e.* with no retrieval), namely GPT-3.5, Llama2-7b-chat, Llama2-13b-chat, and Llama3-8B-instruct (prompts and other details are in App. B.1); (2) standard RAG using the Llama2 family of models (prompts, and other details in App. B.2). All RAG baselines use the same set of retrieved documents.

**Plan-RAG** We use the default configuration of the proposed method, where we use Contriever to retrieve top 10 documents and limit the relevant retrievals to maximum 1 using the relevance expert. We employ the GPT-4o model for generating the query plan (DAG); and the Llama3-instruct$_{8B}$ model for all the other experts in our method.

**Retriever** We use the official Contriever-MS MARCO (Izacard et al., 2022) retriever. It consists of embeddings based on the 2018 English Wikipedia. The Wikipedia articles are segmented into non-overlapping 100-word segments. For all vanilla LLM and RAG baselines, we retrieve 10 documents. We retrieve 10 documents for Plan-RAG as well, but a relevance expert selects the single most relevant document. In the relevance expert $|\mathbf{r}^\star| \geq 1$ setting, it selects all documents deemed relevant.

**Evaluation metrics** We employ three evaluation metrics: Accuracy, F1 score, and LLM-Eval. For accuracy, we consider an answer correct if the predicted answer contains the correct answer. This metric provides a relaxed version of exact-match, allowing for some flexibility in answer phrasing. The F1 score is calculated based on the overlap between the prediction and the true answer, balancing precision and recall. In addition, we utilize the recently developed LLM-Eval method (Lin & Chen, 2023), where we employ GPT-3.5-Turbo to compare the answers. This approach leverages the language understanding capabilities of LLMs for evaluation. Due to computational costs, we apply LLM-Eval to a subset of experimental datasets. The details are discussed in App. D.

### 4.1 MAIN RESULTS

In this section, we present the main results of the experiments in particular we discuss about performance, depth of the reasoning DAG, attribution and evaluation using a large language model.

**Performance** Plan-RAG consistently outperforms vanilla LLM baselines, RAG baselines, and two state-of-the-art RAG frameworks, Self-RAG (Asai et al., 2023) and RQ-RAG (Chan et al., 2024), across three multi-hop datasets: HotpotQA, StrategyQA, and Arc-Challenge. Designed specifically to handle complex, multi-hop queries, Plan-RAG excels in these tasks by effectively using a reasoning DAG and *plug-and-play* experts. Although primarily focused on multi-hop reasoning, Plan-RAG also performs competitively on the single-hop dataset, PopQA, demonstrating that it does not sacrifice performance on simpler queries. The complete results are provided in Table 3.

**Reasoning DAG depth** Table 2 shows the reasoning DAG depths for all the datasets. As expected, the multi-hop query datasets exhibit a DAG depth greater than 1, indicating that multiple atomic queries must be answered at different depths to answer the main query. In contrast, the single-hop dataset typically shows a reasoning DAG depth of 0, which is both expected and desired, as these

Table 3: **Experiment results:** We report two metrics: *Acc*, which measures whether the true answer is a subset of the generated answer, and *F1*, which captures the overlap between the true and generated answers. Plan-RAG achieves the highest performance on both metrics for all multi-hop datasets and demonstrates competitive results on the single-hop dataset.

| | Model | *Multi-hop* | | | | | | *Single-hop* | |
| | | HotpotQA | | StrategyQA | | Arc-Challenge | | PopQA | |
| | | Acc | F1 | Acc | F1 | Acc | F1 | Acc | F1 |
|---|---|---|---|---|---|---|---|---|---|
| Vanilla LLM | GPT-3.5$_{turbo}$ | 27.15 | 37.97 | 55.51 | 31.58 | 66.43 | 78.40 | 27.45 | 28.28 |
| | Llama2-chat$_{7B}$ | 14.50 | 20.68 | 40.25 | 37.65 | 48.90 | 60.53 | 20.70 | 22.30 |
| | Llama2-chat$_{13B}$ | 17.69 | 25.40 | 44.46 | 42.27 | 57.47 | 67.79 | 19.51 | 21.59 |
| | Llama3-instruct$_{8B}$ | 21.35 | 25.57 | 48.42 | 21.24 | 71.40 | 79.99 | 21.23 | 22.62 |
| RAG | Llama2-chat$_{7B}$ | 16.60 | 24.71 | 31.21 | 5.11 | 40.13 | 53.03 | 22.76 | 19.22 |
| | Llama2-chat$_{13B}$ | 18.22 | 27.10 | 43.98 | 29.46 | 53.33 | 64.54 | 29.82 | 25.71 |
| | Llama3-instruct$_{8B}$ | 25.49 | 31.22 | 47.35 | 31.51 | 73.43 | 80.40 | 36.52 | **35.23** |
| SoTA | RQ-RAG | 23.20 | 20.11 | 47.46 | 45.57 | 64.98 | 64.98 | 32.66 | 31.68 |
| | Self-RAG$_{7B}$ | 33.18 | 21.55 | 60.5 | 13.55 | 67.32 | 52.81 | 44.31 | 15.63 |
| | Self-RAG$_{13B}$ | 33.93 | 21.83 | 63.4 | 21.49 | 73.12 | 56.74 | **44.53** | 16.34 |
| | Plan-RAG$_{8B}$ | **35.67** | **39.68** | **69.49** | **64.03** | **74.12** | **81.30** | 36.09 | 35.20 |

simpler queries do not require further decomposition into subqueries. This showcases that Plan-RAG's reasoning DAG effectively adapts its complexity based on the complexity of the query.

**Attribution**   Plan-RAG supports attribution by design, ensuring a one-to-one mapping between generation and the retrieved document. For this, we analyze the frequency of Plan-RAG generating answers with proper attribution to a retrieved passage. Using PopQA, we sample correctly answered queries and check whether generated answer is a substring of the corresponding retrieved document. This setup is inspired by Asai et al. (2023). Our experiment reveal that Plan-RAG correctly attributes 76% of its generations to the retrieved passage (*i.e.*, 76% of the queries have the answer as a substring in the retrieved documents), while 12% of answers come from its world knowledge (*i.e.*, there is no retrieve document). Consequently, only 12% of answers fall outside the retrieved passage (*i.e.*, answer not present as a substring in the retrieve document) or require reasoning over the retrieved data. These results demonstrate that Plan-RAG provides correct attribution or explicitly uses world knowledge at-least 88% of the time, highlighting its strong attribution capability. We further evaluate Plan-RAG using a random sample of 1500 HotpotQA queries and experiment with the configuration employing a relevance expert ($|\mathbf{r}^\star| \geq 1$). In this setup, the expert returns a set of relevant documents rather than a single document, potentially compromising accuracy by losing the one-to-one mapping between retrieved documents and generated answers. However, we observe that due to the atomic nature of the subqueries, the relevance expert selects a single retrieved document 88% of the time. Thus, it maintains a high level of attribution even in this multi-document setting. This finding suggests that Plan-RAG's approach of generating atomic subqueries naturally gravitates towards attribution even when given the option to use multiple documents. We detail the results in Table 5.

**LLM evaluation**   For a more sophisticated evaluation, we employ GPT-3.5-turbo as an external evaluator to compare the answers generated by Self-RAG and Plan-RAG on the PopQA dataset. As shown in Table 4, while the accuracy metrics indicate a performance gap of 8.22% and 8.44% between Plan-RAG$_{8B}$ and Self-RAG$_{7B}$ and Self-RAG$_{13B}$ respectively, the LLM-based evaluation reveals a significantly smaller difference. The LLM-Eval metric shows only a 2.09% difference compared to Self-RAG$_{7B}$ and a 3.32% difference compared to Self-RAG$_{13B}$. This suggests that while Plan-RAG may not always produce generations for accuracy measurements, its generations are often similar to the true answer, as judged by an LLM.

Table 4: **LLM-Eval:** Comparison of Self-RAG and Plan-RAG on the PopQA dataset using Accuracy and the LLM-Eval metric. LLM-Eval reveals that the difference in the output is minimal.

| Metric | Self-RAG$_{7B}$ | Self-RAG$_{13B}$ | Plan-RAG$_{8B}$ | Diff |
|---|---|---|---|---|
| **Accuracy** | 44.31% | 44.53% | 36.09% | 8.22% / 8.44% |
| **LLM-Eval** | 44.83% | 46.06% | 42.74% | 2.09% / 3.32% |

Table 5: **Relevance expert experiment:** Summary of subquery attribution, highlighting the efficiency of the critic expert configuration.

| Metric | Value |
|---|---|
| Total Queries | 1,500 |
| Total Sub-Queries | 3,978 |
| Accuracy (%) | 39.73 |
| Single Doc (%) | 88.5 (3,520) |
| Multiple Docs (%) | 11.5 (458) |

Table 6: **Ablation studies:** Comparison of configurations using 1500 HotpotQA queries to assess performance in various configs.

| Configuration | Accuracy (%) | F1 Score |
|---|---|---|
| Critic Expert | 36.60 | 40.72 |
| Always Retrieve | 37.13 | 41.52 |
| Relevance-Expert | 39.33 | 42.01 |
| No Relevance-Expert | 31.60 | 36.18 |

## 4.2 ABLATION STUDY

We conduct a series of ablation studies to evaluate the effectiveness of key components in the Plan-RAG framework. Specifically, we focus on two critical elements: the *critic expert* and the *relevance expert*. These studies aim to quantify the impact of each component on the overall system performance, measured in terms of accuracy and F1 score.

**Effectiveness of the Critic Expert**   We evaluate the efficacy of the *critic expert* in Plan-RAG using 1500 random queries from HotpotQA. We compare two configurations: (1) a critic expert that dynamically decides whether to trigger retrievals, and (2) a baseline that consistently retrieves after generating $k$-tokens. Both setups are constrained to a single retrieval ($|\mathbf{r}^*|=1$) per subquery. The critic expert configuration achieved an accuracy of $36.60$ and an F1 score of $40.72$, while the baseline attained an accuracy of $37.13$ and an F1 score of $41.52$. Across the 1500 queries (3926 subqueries), the critic expert triggered 2530 retrievals, compared to 3163 in the always-retrieve setup. This represents a 600 reduction in retrievals. Notably, this significant decrease in retrievals resulted in only a marginal performance drop of $0.5\%$ in accuracy and $0.80$ in F1 score. These results demonstrate that the critic expert can improve retrieval efficiency while maintaining near-equivalent performance.

**Effectiveness of the Relevance Expert**   We evaluate the *relevance expert* in the Plan-RAG framework using 1500 random queries from HotpotQA. Two configurations are compared: (1) with a relevance expert, and (2) a no relevance expert using all retrievals. Both setups initially retrieved 10 documents per subquery. The expert configuration significantly outperformed the no relevance expert, achieving 39.33 accuracy and 42.01 F1 score, compared to 31.6 accuracy and 36.18 F1 score for the no relevance expert configuration. The performance gap, despite the second configuration using more documents, can be attributed to the reasoning limitations of the Llama3-8B model and the negative impact of noisy retrievals on generation. These results highlight the crucial role of the relevance expert in enhancing performance by effectively filtering and providing only relevant retrievals.

## 5 DISCUSSION AND CONCLUSION

In this paper, we present Planning-guided Retrieval Augmented Generation (Plan-RAG), a novel framework designed to tackle critical challenges in Retrieval-Augmented Generation (RAG), addressing performance and hallucinations in complex queries, and lack of attribution. Unlike traditional RAG systems that follow a *retrieve-then-reason* approach, Plan-RAG shifts to a *plan-then-retrieve* paradigm. This shift facilitates a more structured, efficient, and interpretable methodology for managing complex queries. Key innovations of Plan-RAG include a reasoning plan represented as a DAG and a set of *plug-and-play* experts. These advancements yield significant improvements across various critical dimensions like performance, attribution, as evidenced by the experimental results. The *plug-and-play* experts allow for seamless substitution with alternative or emerging language models, ensuring that Plan-RAG remains adaptable to evolving technologies. Furthermore, the framework's design enables easy integration with various language models without necessitating fine-tuning, establishing it as a versatile and practical solution. Future work will focus on incorporating additional experts capable of early exiting from the reasoning DAG, developing specialized experts for mathematical calculations and logical reasoning, and implementing dynamic expert allocation conditioned on the input query.

**Reproducibility statement**   We will open-source the code for Plan-RAG upon acceptance. We provide all the expert prompts in the Appendix and use the official code for Self-RAG and RQ-RAG.

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

APPENDICES

This supplementary document is organized as follows: App. A discusses various datasets and their specific characteristic used in the experiments. App. B describes the baselines models, their setup and specific prompts that are used. This section is further divided into two subsections: App. B.1 and App. B.2 for vanilla LLM and RAG baselines respectively. In App. C, we discuss the experiment details and hyperparameters used by the proposed as well as the competitive methods. In App. D we discuss the setup and details of the LLM-Eval experiment setup and App. E discusses various ablation studies and other experiments.

# A   DATASET DETAILS

In this section, we discuss the datasets used in the experiments. The datasets are particularly characterize into multi-hop and single-hop depending on the nature of the queries they contain.

## A.1   MULTI-HOP QA

For multi-hop queries, we focus on the three datasets: HotpotQA (Yang et al., 2018), StrategyQA (Geva et al., 2021), and Arc-Challenge (Clark et al., 2018).

**HotpotQA**   It is a multi-hop datasets from Wikipedia. The questions are diverse and not constrained to any pre-existing knowledge bases or knowledge schemas. HotpotQA is a question-answering dataset collected on the English Wikipedia, containing 7405 total queries in the *dev-fullwiki* setup. Although, each question in the dataset comes with two gold paragraphs, as well as a list of sentences in these paragraphs that crowd workers identify as supporting facts necessary to answer the question, we in the experiments do not use them and use contriever to fetch the relevant documents.

**StrategyQA**   It is a question-answering benchmark where multiple reasoning steps are required in order to answer the question. Also, the answer should be inferred using a strategy. Questions are short, topic-diverse, and cover a wide range of strategies. We use the dataset that is available on the Self-RAG repository Asai et al. (2023). The dataset consists of 2,234 question-answer pairs, each consisting of a strategy question.

**Arc-Challenge**   It is a multiple-choice reasoning dataset created from scientific exam. It is a subset of the broader ARC (AI2 Reasoning Challenge) dataset and is considered more challenging due to the inclusion of harder questions that often require external knowledge. We use the dataset that is available on the Self-RAG repository Asai et al. (2023). The dataset consists of total 1095 queries.

## A.2   SINGLE-HOP QA

To judge the models performance on Single-Hop queries we use PopQA (Mihaylov et al., 2018).

**PopQA**   PopQA is an open-domain question-answering dataset designed to assess a model's ability to retrieve and generate answers based on factual knowledge. The dataset consists of factual questions, many of which require specific knowledge of popular culture, history, and general world facts. In total the size of the dataset is 1399 question-answer pairs.

# B   BASELINES

In this section, we describe the baseline models employed in our experiments, which include both vanilla LLMs and retrieval-augmented generation (RAG) models. We outline the models, setup, and hyperparameters used for each configuration. In App. B.1, we discuss the vanilla LLM models and in App. B.2 we discuss the RAG models. The model include GPT-3.5-turbo, Llama2-7B-chat, Llama2-13B-chat, and Llama3-8B-instruct. For all the baseline models we set temperature to 0.0.

## B.1   VANILLA LLM

For vanilla LLM baselines, we use GPT-3.5-turbo, Llama2-7B-chat, Llama2-13B-chat, and Llama3-8B-instruct. The Llama2 and Llama3 models are obtained via HuggingFace and *vLLM* Python

package is used for inference. GPT-3.5-Turbo model is accessed via official API. The prompt for GPT3.5-turbo model is:

> Be precise and give answer to the query. Response should be a valid JSON, that can be passed to "json.loads" directly, with a key as Response which ONLY has 2-3 words. DO NOT use complete sentences or punctuation. In JSON, put every value as a string always, not float.
> Example:
> Query: "What is the capital of France?"
> {"Response": "Paris"}
> Query: "How do you make coffee?"
> {"Response": "Brew ground beans"}
> Now, answer the query: '{*query*}'

The prompt for the Vanilla LLM Llama2 and Llama3 model is:

> ⟨s⟩ [INST] ⟨⟨SYS⟩⟩ You are a concise answering assistant. Follow these rules strictly:
> 1. Respond ONLY to the given QUERY.
> 2. Your entire response must be a valid JSON object.
> 3. The JSON object must have only one key: "Response".
> 4. The value of "Response" must be 2-3 words maximum.
> 5. Do not use complete sentences or punctuation in the "Response" value.
> 6. Ensure the JSON can be directly parsed by json.loads().
> 7. In JSON, put every value as a string always, not float.
> Examples:
> Query: What is the capital of France?
> {"Response": "Paris"}
> Query: How do you make coffee?
> {"Response": "Brew ground beans"}
>
> NOTE: Always respond with ONLY the JSON object, nothing else.⟨⟨/SYS⟩⟩
> Now, answer the query: '{*query*}' [/INST]

## B.2 RAG BASELINES

For RAG baselines, we use Llama2-7B-chat, Llama2-13B-chat, and Llama3-8B-instruct. The Llama2 and Llama3 models are obtained via HuggingFace and *vLLM* Python package is used for inference. For all the RAG baselines we use the Contriever to retrieve top 10 documents conditioned on the query. We retrieve and use the same set of documents for all the baselines.

The prompt for RAG Llama2 and Llama3 model is:

> ⟨s⟩ [INST] ⟨⟨SYS⟩⟩ You are a concise answering assistant. Use the Retrievals while generating the answer and keep the answer grounded in the retrievals. Generate a JSON with a single key "Response" and a value that is a short phrase or a few words. In JSON, put every value as a string always, not float.
>
> NOTE: Generate only JSON without any explanation . Example:
>
> Input: The query is "What is the capital of France?"
> Retrievals are [["Paris is the capital of France."]]
> Generation: {"Response": "Paris"}
>
> Input: The query is "How do you make coffee?"
> Retrievals are [["Brew ground beans are used to make coffee."]]
> Generation: {"Response": "Brew ground beans"} ⟨⟨/SYS⟩⟩

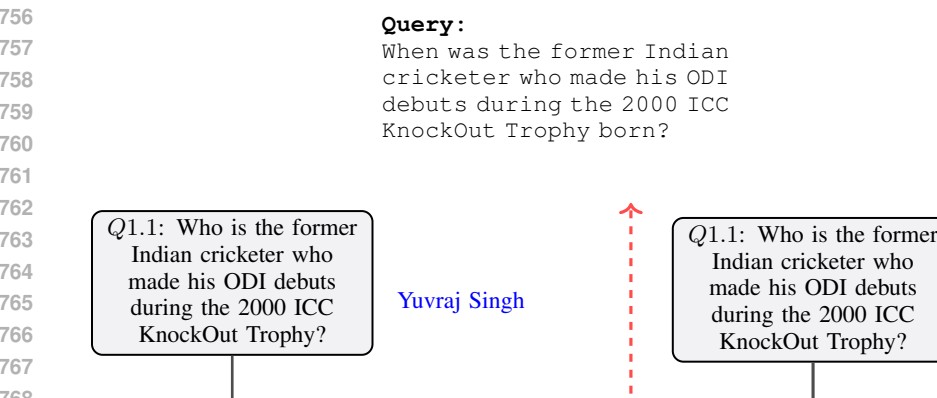

Figure 4: **Plan-RAG Backtracking and Debuggability:** An example from the HotpotQA dataset where Plan-RAG initially produces an incorrect output due to ambiguity in the main query. By backtracking through the reasoning DAG and identifying the source of the error, a small additional context enables Plan-RAG to generate the correct output.

## C  EXPERIMENT DETAILS

In this section, we discuss the setup and details of the two competitive currently SoTA methods Asai et al. (2023) and Chan et al. (2024) as well as the proposed Plan-RAG method.

### C.1  COMPETITIVE METHOD DETAILS

**Self-RAG (Asai et al., 2023)**: It is an open source framework wherein the base Llama2 models are trained to learn special reflection tokens. The reflection tokens are then used to judge the requirement for retrievals, relevance of the retrieved documents and the accuracy of the output. We tested Self-RAG$_{7b}$ and Self-RAG$_{13b}$ on both single and multi-hop datasets, judging it using F1, subset match as well as LLM-Eval. We use 10 documents as context; these are retrieved using contriever at the beginning of each query iteration. The temperature was set to zero to maintain non-stochasticity.

**RQ-RAG (Chan et al., 2024)** It is an open source framework where in the base Llama2 model is trained to enable it to dynamically refine search queries through rewriting, decomposing, and clarifying ambiguities. Control tokens are used to direct the generation process. Furthermore, they use three different sampling methods which includes selection based on perplexity (PPL), confidence, and an ensemble approach, in order to select the final answer. A total of three documents are retrieved at each depth for any given query and the maximum depth is set to 2. Both F1 and subset match are used to judge the models accuracy on single and multi-hop datasets.

### C.2  PLAN-RAG

**Reasoning DAG**    The reasoning DAGs are created by prompting the language model with the query along with several contextual examples. We ensure that the generated DAG is as simple as possible, adhering to the principle that the answers to sub-queries depend solely on their respective parent nodes. Indexed input tags are employed to denote the answers, and these tags are incorporated within sub-queries to clearly illustrate the dependency on their parent nodes.

The prompt used for the DAG generation is:

> You are a reasoning DAG generator expert. The goal is to make a reasoning DAG with minimum nodes.

Given a query, if it is complex and requires a reasoning plan, split it into smaller, independent, and individual subqueries. The query and subqueries are used to construct a rooted DAG so make sure there are NO cycles and all nodes are connected, there is only one leaf node with a single root and one sink. DAG incorporates Markovian property i.e. you only need the answer of the parent to answer the subquery. The main query should be the parent node of the initial set of subatomic queries such that the DAG starts with it. Return a Python list of tuples of parent query and the subatomic query which can be directly given to eval().

For the subquery generation, input a tag ⟨A⟩ where the answer of the parent query should come to make the query complete.
Note: make the dag connected and a rooted tree. for simple queries return the original query only without any reasoning dag.
Examples:

Query: Who is the current PM of India?
DAG: "Q: Who is the current PM of India?"

Query: What is the tallest mountain in the world and how tall is it?
DAG: [("Q: What is the tallest mountain in the world and how tall is it?", "Q1.1: What is the tallest mountain in the world?"), ("Q1.1: What is the tallest mountain in the world?", "Q2.1: How tall is ⟨A1.1⟩?")]

Query: What percentage of the worlds population lives in urban areas?
DAG: [("Q: What percentage of the worlds population lives in urban areas?", "Q1.1: What is the total world population?"), ("Q: What percentage of the worlds population lives in urban areas?", "Q1.2: What is the total population living in urban areas worldwide?"), ("Q1.1: What is the total world population?", "Q2.1: Calculate the percentage living in urban areas worldwide when total population is ⟨A1.1⟩ and population living in urban areas is ⟨A1.2⟩?"), ("Q1.2: What is the total population living in urban areas worldwide?", "Q2.1: Calculate the percentage living in urban areas worldwide when total population is ⟨A1.1⟩ and population living in urban areas is ⟨A1.2⟩?")]

NOTE: Always respond with the JSON object.

**Relevance Expert**

The prompt provided to the relevance expert LM is:

⟨s⟩ [INST] ⟨⟨SYS⟩⟩ You will be provided with a query, along with retrievals and possibly some generation. Your job is to determine if the retrievals are relevant to the query and the generation, and provide useful information to answer the query or not. If the retrievals meet this requirement, respond with the retrieval id that is highly relevant (only one); otherwise, respond with '[No]'.

Example:
Query: Did Snoop Dogg refuse to make music with rival gang members?
Generation:
Retrievals:

1 Calvin Cordozar Broadus Jr. (born October 20, 1971), known professionally as Snoop Dogg (previously Snoop Doggy Dogg and briefly Snoop Lion), is an American rapper, media personality, and actor.

2 Broadus' debut studio album, Doggystyle (1993), produced by Dr. Dre, was released by Death Row Records and debuted at number one on the Billboard 200.

3 In 1993, Broadus was charged with first-degree murder for the shooting of a member of a rival gang who was actually killed by Snoop's bodyguard. Broadus was acquitted

on February 20, 1996.

4 While recording Doggystyle in August 1993, Broadus was arrested and charged with first-degree murder in connection with the shooting death of Philip Woldermariam, a member of a rival gang, who was actually killed by Broadus' bodyguard, McKinley Lee, aka Malik.

5 In 2002, he released the album Paid tha Cost to Be da Bo, on Priority/Capitol/EMI, selling over 1,310,000 copies. The album featured the hit singles 'From tha Chuuuch to da Palace' and 'Beautiful', featuring guest vocals by Pharrell.

Output: [No]

Query: Who is the mother of the director of the film Polish-Russian War (Film)?

Generation: The director of the film Polish-Russian War (Film) is Xawery ˙Zuławski. His parents are

Retrievals:

1 Polish-Russian War (Wojna polsko-ruska) is a 2009 Polish film directed by Xawery ˙Zuławski based on the novel Polish-Russian War under the white-red flag by Dorota Masłowska.

2 Xawery ˙Zuławski (born 22 December 1971 in Warsaw) is a Polish film director. In 1995 he graduated from the National Film School in Łód´z. He is the son of actress Małgorzata Braunek and director Andrzej ˙Zuławski.

3 After an argument in a bar owned by "Left" (Michał Czernecki), 'Strong' meets a 'Gothgirl' Angelica (Maria Strzelecka) at night, an aspiring poet dressed in black, also a virgin and pessimist, for whom 'suicide is a piece of cake'.

4 'Strong' follows Magda. He turns up at the town festival, where she takes part in a miss competition. He cannot reach her, but instead he meets a volunteer, Ala, a girl of his friend Casper, coming from a good family, with whom he spends the afternoon.

5 Production: The film was shot between May 6 and 18, June 2008, in locations of Warsaw, Wejherowo, Sopot, and Gdynia outskirts. The film premiered on

Output: [2]

**Relevance Expert** $|r^\star| \geq 1$

The prompt provided to the multi-relevance expert LM is:

⟨s⟩ [INST] ⟨⟨SYS⟩⟩ You will be provided with a query, along with retrievals and possibly some generation. Your job is to determine if the retrievals are relevant to the query and the generation, and provide useful information to answer the query or not. If the retrievals meet this requirement, respond with the retrieval id that is highly relevant (only one); otherwise, respond with '[No]'.

Example:
Query: Did Snoop Dogg refuse to make music with rival gang members?
Generation:
Retrievals:

1 Calvin Cordozar Broadus Jr. (born October 20, 1971), known professionally as Snoop Dogg (previously Snoop Doggy Dogg and briefly Snoop Lion), is an American

rapper, media personality, and actor.

2 Broadus' debut studio album, Doggystyle (1993), produced by Dr. Dre, was released by Death Row Records and debuted at number one on the Billboard 200.

3 In 1993, Broadus was charged with first-degree murder for the shooting of a member of a rival gang who was actually killed by Snoop's bodyguard. Broadus was acquitted on February 20, 1996.

4 While recording Doggystyle in August 1993, Broadus was arrested and charged with first-degree murder in connection with the shooting death of Philip Woldermariam, a member of a rival gang, who was actually killed by Broadus' bodyguard, McKinley Lee, aka Malik.

5 In 2002, he released the album Paid tha Cost to Be da Bo, on Priority/Capitol/EMI, selling over 1,310,000 copies. The album featured the hit singles 'From tha Chuuuch to da Palace' and 'Beautiful', featuring guest vocals by Pharrell.

Output: [No]

Query: Who is the mother of the director of the film Polish-Russian War (Film)?
Generation: The director of the film Polish-Russian War (Film) is Xawery Żuławski. His parents are
Retrievals:

1 Polish-Russian War (Wojna polsko-ruska) is a 2009 Polish film directed by Xawery Żuławski based on the novel Polish-Russian War under the white-red flag by Dorota Masłowska.

2 Xawery Żuławski (born 22 December 1971 in Warsaw) is a Polish film director. In 1995 he graduated from the National Film School in Łódź. He is the son of actress Małgorzata Braunek and director Andrzej Żuławski.

3 After an argument in a bar owned by "Left" (Michał Czernecki), 'Strong' meets a 'Gothgirl' Angelica (Maria Strzelecka) at night, an aspiring poet dressed in black, also a virgin and pessimist, for whom 'suicide is a piece of cake'.

4 'Strong' follows Magda. He turns up at the town festival, where she takes part in a miss competition. He cannot reach her, but instead he meets a volunteer, Ala, a girl of his friend Casper, coming from a good family, with whom he spends the afternoon.

5 Production: The film was shot between May 6 and 18, June 2008, in locations of Warsaw, Wejherowo, Sopot, and Gdynia outskirts. The film premiered on

Output: [2]

Query: What is the capital of Australia and when did it become the capital?
Generation:
Retrievals:

1 Canberra is the capital city of Australia. It was officially named the capital in 1913, after the site was chosen as a compromise between rivals Sydney and Melbourne. The city was designed by American architects Walter Burley Griffin and Marion Mahony Griffin, who won an international design competition.

> 2 The Great Barrier Reef, located off the coast of Queensland in northeastern Australia, is the world's largest coral reef system. It is composed of over 2,900 individual reefs and 900 islands stretching for over 2,300 kilometers. The reef is home to diverse marine life and is visible from outer space.
>
> 3 Prior to Canberra becoming the capital, Melbourne served as the temporary seat of government from 1901 to 1927. The Parliament of Australia was officially opened in Canberra on 9 May 1927, marking the city's true beginning as the nation's capital.
>
> Output: [1],[3]

**Critic Expert**

The prompt provided to the critic expert LM is:

> You will be provided with a query, generation, and evidence (optional). Your task is to determine whether the information in the generation can be fully verified by the evidence (if present) or if it requires external verification. If the generation can be verified solely with the evidence (if present), output False. If additional information is needed to verify the generation, output True.
> NOTE: If the generation mentions that it is not sure about the answer or does not have the resources to answer, output True.
>
> Example:
>
> Query: Explain the use of word embeddings in Natural Language Processing.
> Evidence: Word embedding is the collective name for a set of language modeling and feature learning techniques in natural language processing (NLP) where words or phrases from the vocabulary are mapped to vectors of real numbers. Conceptually it involves a mathematical embedding from a space with one dimension per word to a continuous vector space with a much lower dimension.
> Generation: Word embeddings are useful for tasks such as sentiment analysis, text classification, predicting the next word in a sequence, and understanding synonyms and analogies.
> Output: True
>
>
> Query: What is the capital of France?
> Evidence: Paris is the capital and most populous city of France. Situated on the Seine River, in the north of the country. Generation: The capital of France is Paris.
> Output: False
>
>
> Query: Find the area of a circle given its radius. Radius = 4
> Evidence:
> Generation: The area of the circle is
> Output: False

# D  LLM-EVAL

For a more sophisticated evaluation of Plan-RAG performance on PopQA dataset, we leverage the LLM-Eval framework, as introduced by Lin & Chen (2023), where we utilize GPT-3.5-turbo as an external evaluator to compare the answers generated by Self-RAG (Asai et al., 2023) and Plan-RAG. While traditional accuracy metrics reveal a performance gap of 8.22% between Plan-RAG$_{8B}$ and Self-RAG$_{7B}$, and 8.44% between Plan-RAG$_{8B}$ and Self-RAG$_{13B}$—the LLM-Eval metric shows a different picture. As shown in Table 4, the LLM-based evaluation indicates a much smaller gap, with only a 2.09% difference compared to Self-RAG$_{7B}$ and a 3.32% difference compared to Self-RAG$_{13B}$. This suggests that although Plan-RAG may not always generate outputs that improve accuracy scores, its responses are often highly similar to the correct answers when evaluated through

the lens of an LLM model. Due to the computational constraints, we apply LLM-Eval only on PopQA dataset.

The prompt for LLM-Eval using GPT-3.5-Turbo model is:

> You are a judge of if two answers (ANSWER and PREDICTED) of the QUESTION aligns or not with each other. To determine if two answers align, compare their content while disregarding differences like punctuation or formatting. Focus on the core factual information they convey. If the essence of both answers is consistent, despite slight variations in wording, classify them as 'Correct.' However, if there are substantial differences in the factual information presented, classify them as 'Incorrect.'
>
> Please do not use any other words except Correct or Incorrect

## E  ABLATION STUDIES AND OTHER EXPERIMENTS

In this section, we discuss the setup and other details about the various experiments and ablation studies performed.

### E.1  RELEVANCE EXPERT

We conducted experiments on 1500 randomly selected queries from the HotpotQA dataset, retrieving $k{=}10$ documents per query and thereafter using the relevance expert ($|\mathbf{r}^\star| \geq 1$) to get the set of relevant documents $\mathbf{r}^\star$. The relevance expert returned a set of relevant retrievals, and the goal is to observe how often the retriever selected only one document due to the atomic nature of the subqueries. In these cases, relevant information for each subquery tends to reside in a single document.

Our findings reveal that $88.5\%$ of the subqueries retrieved at most one relevant document ($|\mathbf{r}^\star| \leq 1$), while only $11.5\%$ retrieved more than one document ($|\mathbf{r}^\star| > 1$). This demonstrates that even when multiple documents are available, the majority of generations maintain a one-to-one mapping with a document, preserving attribution. Detailed statistics are shown in Table 5.

### E.2  EFFECTIVENESS OF THE CRITIC EXPERT

We conduct an experiment on 1500 randomly selected queries from the HotpotQA dataset, comparing two configurations: one where the critic expert triggers retrievals, and another where retrieval occurs after every $k$ tokens. In both setups, we retrieve 10 documents using the Contriever retriever, and the relevance expert selects the most relevant document, *i.e.* $|r^\star| = 1$. The objective is to demonstrate the effectiveness of the critic expert by showing that it reduces the number of retrievals while maintaining similar accuracy.

Across the 1,500 queries, there were 3,926 subqueries. The always-retrieve setup used a total of 3,163 retrievals, whereas the critic expert setup used 2,530 retrievals. The critic expert configuration achieved an accuracy of 36.60 and an F1 score of 40.72, while the always-retrieve configuration achieved an accuracy of 37.13 and an F1 score of 41.52. This represents a substantial reduction of 600 retrievals, with only a minor performance drop of 0.5% in accuracy and 0.80 in F1 score. These results highlight that the critic expert can significantly enhance retrieval efficiency while maintaining nearly equivalent performance. Detailed statistics are provided in Table 6. For both the experiments the default version of Plan-RAG was used with GPT-4o call for generating the reasoning DAG and all the experts were Llama3-8B-instruct model.

### E.3  EFFECTIVENESS OF THE RELEVANCE EXPERT

We conduct an experiment on 1500 randomly selected queries from the HotpotQA dataset, comparing two configurations: (1) one where the relevance expert filters retrievals, and (2) one without the expert, using all retrieved documents. In both setups, 10 documents are retrieved using the Contriever retriever. The goal of the experiment is to demonstrate the effectiveness of the relevance by showing that it filters out noisy retrievals, thereby enabling the generator to reason more effectively and produce relevant answers.

The relevance expert configuration significantly outperformed the setup without the expert, achieving an accuracy of 39.33 and an F1 score of 42.01, compared to 31.6 accuracy and 36.18 F1 for the latter. Despite the no-expert configuration using more documents, its performance deteriorated due to the reasoning limitations of the Llama3-8B model and the negative impact of noisy retrievals on generation. These results highlight the critical role of the relevance expert in enhancing system performance by filtering and prioritizing relevant retrievals. Detailed statistics are provided in Table 6. For both the experiments the default version of Plan-RAG was used with GPT-4o call for generating the reasoning DAG and all the experts were Llama3-8B-instruct model.

