# OpenReview forum: "Plan-RAG: Planning-guided Retrieval Augmented Generation"
_ICLR.cc/2025/Conference — Submitted to ICLR 2025_

### Official Review · Reviewer_RUfe · 2024-10-20

**Soundness:** 3
**Presentation:** 2
**Contribution:** 3
**Rating:** 6
**Confidence:** 3

**Summary:**

The paper introduces Plan-RAG, a framework that shifts from the traditional “retrieve-then-reason” paradigm used to a plan-then-retrieve paradigm. Plan-RAG decomposes queries into atomic sub-queries using a reasoning Directed Acyclic Graph, which allows parallelized retrieval and generation. The system utilizes frozen language models as plug-and-play experts to guide retrieval and generation. This approach significantly reduces hallucinations, improves attribution, and enhances efficiency, especially for multi-hop reasoning tasks.

**Strengths:**

1. Plan-RAG offers a novel “plan-then-retrieve” paradigm with a reasoning DAG, providing a new way to handle complex queries in retrieval-augmented generation.
2. The framework is technically sound and evaluated across multiple datasets, demonstrating improvements in efficiency, hallucination reduction, and attribution.
3. The key concepts, particularly the reasoning DAG and plug-and-play expert system, are clearly presented with helpful examples and diagrams.
4. Plan-RAG addresses important challenges in hallucination and attribution, offering a scalable solution that works with frozen language models.

**Weaknesses:**

1. This paper lacks of the analysis on the DAG generation part. It remains unclear how does DAG performs vs. a plan without DAG structure. You could conduct an ablation study comparing Plan-RAG with and without the DAG structure.
2. This paper lacks of the error analysis about the results. For example, which module leads to the most errors and has the greatest impact on the model results. You could provide a breakdown of errors by different experts and to quantify the impact of each expert on overall performance.
3. The paper only does experiments on an 8B model, which lacks of generalization experiments on different sizes models. How does the performance and efficiency of Plan-RAG scale with model size (e.g., GPT-4)?

**Questions:**

Due to the application of multiple experts, you may provide concrete latency measurements comparing Plan-RAG to baseline methods for an analysis of how latency scales with query complexity. It would be better if the authors could discuss potential optimizations to reduce overhead in real-world deployments.

---

> ### Author Response · Authors · 2024-11-19
>
> We thank the reviewer for their comments. We address the concerns below.
>
> **(W1) Ablation without DAG:** The reasoning DAG is central to Plan-RAG, providing features like parallelizable queries, structured information flow, and attribution. Removing the DAG would undermine the core framework and its intended contributions. However, we do include ablations with the plug-and-play experts in Section 4.2 to illustrate their individual impacts within the Plan-RAG framework.
>
> **(W2) Error analysis of each expert:** Thank you for the suggestion regarding expert-level error analysis. Since the experts are sequentially connected, any error in one component can cascade to subsequent ones, complicating isolated error analysis. This setup presents challenges in accurately attributing errors to individual experts based on final output. We acknowledge the importance of this analysis and will consider it as future work.
>
> **(W3) Plan-RAG with GPT4o as expert:** We evaluate Plan-RAG with GPT-4o as the expert on a subset of the HotpotQA dataset (1000 queries). The results are detailed in the table below.
>
>
> | HotpotQA                | Accuracy | F1     |
> | --------                | -------- | ------ |
> | Vanilla LLM (GPT-4o)    | 40.69    | 48.97  |
> | $\text{Plan-RAG}_{\text{GPT-4o}}$        | 50.75    | 54.16  |
>
>
> **(Q) Latency due to multiple experts and how it could be optimized:** We recognize that employing multiple experts introduces some latency; however, this is also true for other competitive state-of-the-art RAG methods that utilize multiple LLMs or LLM calls. In contrast, Plan-RAG’s reasoning DAG structure allows for parallelization, which mitigates overall latency. Section 3.2 discusses the complexity of Plan-RAG, and we will expand the manuscript with further details on this topic.

---

> > ### Comment · Reviewer_RUfe · 2024-11-19
> >
> > Thanks for your reply. I will keep my scores.

---

### Official Review · Reviewer_FQd4 · 2024-10-22

**Soundness:** 2
**Presentation:** 3
**Contribution:** 2
**Rating:** 3
**Confidence:** 4

**Summary:**

This paper introduces Plan-RAG, a *plan-then-retrieve* RAG framework that models the RAG process as a DAG, decomposing queries into subqueries and uses various expert llms to decide whether to retrieve and ensure the relevance of the retrieved documents.

In experiments, the paper compares the performance of Plan-RAG with recent SOTA methods, Self-RAG and RQ-RAG, on four QA datasets. The results demonstrate the effectiveness of the proposed method to some extent. In the ablation study, the paper emphasizes the importance of the critic and relevance expert designed in the Plan-RAG framework.

**Strengths:**

- The motivation of the paper is clear and the framework is easy to understand.
- The proposed framework can achieve superior performance compared to various baselines without fine-tuning the LLM.

**Weaknesses:**

1. There are some inconsistencies in the equations and symbols in this paper. For example, the symbol $v$ in eq (1) and on line 211 is not defined; the author might have meant $q$ instead? Also, on line 238, should $q$ be $\tilde{q}$?
2. The novelty of this paper needs further verification. Firstly, the "plan-then-retrieve" framework is not introduced for the first time in this paper; a very similar work can be found in [1]. Additionally, this paper's approach shares similarities with methods based on tool planning (viewing retrieval as a tool) and the multi-agent RAG framework. Moreover, methods related to query rewrite [2][3] and document relevance judgment [4] exist.

3. There are unfair settings in the baselines of Table 3. (1) In Plan-RAG, the most challenging planning process for llm is generated by GPT-4o, which is unfair for the standard RAG baseline. How would it be if planning were generated by Llama-8b or if GPT-4o were used as the model for standard RAG? (2) The baselines in the vanilla LLM are relatively weak; testing the effects of stronger models like GPT-4 or GPT-4o or Claude-3 would be more persuasive.

4. The performance of this framework on single-hop data seems unsatisfactory.

5. Two important benefits of Plan-RAG are efficiency and debuggability. However, the paper lacks relevant experiments to prove these points, and there seems to be no manifestation of debuggability in the methods section.

[1] PlanRAG: A Plan-then-Retrieval Augmented Generation for Generative Large Language Models as Decision Makers, NAACL 2024

[2] Query Rewriting for Retrieval-Augmented Large Language Models, EMNLP 2023

[3] RaFe: Ranking Feedback Improves Query Rewriting for RAG, Findings of EMNLP 2024

[4] Self-RAG: Learning to Retrieve, Generate, and Critique through Self-Reflection, NeurIPS 2023

**Questions:**

1. Why does Plan-RAG perform poorly on single-hop tasks?
2. A term that consistently appears in this paper is "attribution," which is not explained in the paper in the context of the RAG system. How should this term be understood? I can only roughly infer its relevance to the granularity of subqueries from line 400. Is this understanding correct?
3. All steps of planning in this paper seem to be generated in a one-pass manner. In fact, another paradigm for planning involves iterative generation at each step (e.g. ReAct). How should these two paradigms be viewed in terms of the advantages and disadvantages for planning effectiveness?

---

> ### Author Response · Authors · 2024-11-19
>
> We thank the reviewer for their comments. We address the concerns below.
>
> **(W1) Minor Inconsistencies:** We will fix them in the manuscript.
>
> **(W2) Novelty of "*plan-then-retrieve*":** While we acknowledge that the *"plan-then-retrieve"* framework has been explored in previous research, such as PlanRAG[NAACL 2024], our approach introduces several key innovations that distinguish it from existing methods. Specifically, we propose the use of a directed acyclic graph (DAG) for structured reasoning, which offers numerous benefits. This DAG structure allows for the decomposition of complex queries into interrelated sub-queries, enabling parallelized retrieval and generation. This structured approach not only improves efficiency but also reduces hallucinations by ensuring each sub-query is contextually linked. In contrast, other methods may mention planning but lack a proper structure, resulting in mere decomposition rather than a true plan. Additionally, the use of subatomic sub-queries ensures a one-to-one mapping between subqueries and retrieved documents, enhancing attribution.
>
> While we recognize the similarities with methods based on tool planning, multi-agent RAG frameworks, query rewriting, and document relevance judgment, the proposed framework's unique contributions lie in its structured reasoning, parallelized subatomic subqueries, relevant flow of information, and enhanced attribution mechanisms.
>
> **(W3) Powerful baselines:** We add below the details of the baselines on a subset of HotpotQA dataset (1000 queries):
>
> | HotpotQA                 | Accuracy | F1       |
> | --------                 | -------- | -------- |
> | Vanilla LLM (GPT-4o)     | 40.69    | 48.97    |
> | Vanilla RAG (GPT-4o)     | 41.10    | 47.11    |
> | $\text{Plan-RAG}_{\text{GPT-4o}}$       | 50.75    | 54.16    |
>
>
> **(W4) Performance on single-hop:** Plan-RAG is designed primarily for multi-hop queries and its strength lies in generating a reasoning DAG. For single-hop queries, generally there is no plan required and thus not well-suited for the simplicity of single-hop queries. However, in section 4.1 we perform LLM evaluation and showcase that the difference between Self-RAG~13B~ and Plan-RAG~8B~ is 3.3% and not 8.4% according to the accuracy metric (see Table 4 in the manuscript).
>
> **(W5) Experiment on efficiency and debuggability:** For debuggability, the structured reasoning provided by the DAG allows for clear mapping between subquery, generation, and retrieved document. This mapping facilitates easier identification and correction of errors, as each part of the generated response can be traced back to a specific source. While the current manuscript may not explicitly detail debuggability, the inherent design of Plan-RAG supports this benefit. We have an example of this in the appendix, Figure 4. We plan to explore this further in future work.
>
> **(Q2) Attribution in RAG:** In the context of Plan-RAG, "attribution" refers to the ability to trace the generated responses back to specific retrieved documents. This is crucial for ensuring the reliability and transparency of the information provided by the system.
>
> Plan-RAG achieves enhanced attribution through its structured reasoning process using a DAG. This structure decomposes complex queries into interrelated atomic sub-queries. Each sub-query typically retrieves a single document, creating a clear one-to-one mapping between the sub-query and the retrieved document. This granularity ensures that each part of the generated response can be directly linked to a specific source, making it easier to verify the information and reducing the risk of hallucinations.
>
> **(Q3) Comparison against multiple pass planning (*e.g.* ReAct):** ReAct and Plan-RAG are both designed to enhance LLMs performance, but they differ significantly in their approaches. ReAct combines reasoning and acting in an iterative loop, allowing the model to refine its understanding and actions dynamically, making it highly adaptable to evolving tasks. In contrast, Plan-RAG uses a directed acyclic graph (DAG) to structure the reasoning process, decomposing complex queries into interrelated sub-queries for parallelized retrieval and generation. This structured approach improves efficiency and ensures clear attribution by linking each sub-query to a specific document, reducing hallucinations. While ReAct excels in dynamic decision-making scenarios, Plan-RAG is optimized for efficiently handling complex, multi-hop queries with enhanced transparency and reliability.

---

> > ### Comment · Reviewer_FQd4 · 2024-11-21
> > **Reply to author's rebuttal**
> >
> > Thanks for the authors' response, which has addressed some of my concerns. However, my worries about the novelty persist. And I have some additional questions:
> >
> > **Regarding W3:**
> >
> > I notice in Table 3 of the paper that vanilla GPT-3.5 can achieve performance comparable to Plan-RAG (37.97 vs 39.68 on F1). Why is it that in the authors' supplementary experiments, the performance of vanilla GPT-4o is significantly lower than Plan-RAG (48.97 vs 54.16 on F1), which contradicts our expectations (GPT-4o should be much stronger than GPT-3.5)? Furthermore, the performance of Plan-RAG is inconsistent with Table 3. Additionally, why does the F1 performance of vanilla RAG GPT-4o fall short compared to vanilla GPT-4o? This seems also contradictory to our common expectations. Why does adding RAG have a negative impact on GPT-4o?
> >
> > **Regarding Q3:**
> > I do not agree with the authors' explanation. ReAct is also proposed for handling complex planning problems. Apart from the efficiency aspect mentioned by the authors (which has not been empirically validated in the paper), I am skeptical that ReAct's performance might not be worse than the method proposed in this paper.
> >
> > If the above issues can be addressed, I would be happy to reconsider my score. However, if not, I believe there is still substantial room for improvement in this paper.

---

> > > ### Author Response · Authors · 2024-11-25
> > >
> > > Thank you for the prompt response! We address the additional questions below:
> > >
> > > **Explanation regarding W3:** We would like to point out that the additional experiments in the supplementary material are conducted on a subset of the HotpotQA dataset (1000 queries, as specified in the previous response). This could explain the observed inconsistencies with Table 3 which is on the full dataset. Additionally, the F1 score measures the overlap between the predicted output and the true output. Sometimes, the LLM generates a long sentence that contains the right answer, resulting in a lower F1 score, e.g., "The name of the author is Mark" vs. "Mark". Therefore, the primary metric for evaluation in the manuscript is Accuracy, with the F1 score serving as a secondary metric.
> > >
> > > **ReAct Comparison:** We perform an experiment on a subset of the HotpotQA dataset (1000 subqueries). The results are as follows:
> > >
> > > | Method                 | Accuracy |
> > > | --------               | -------- |
> > > | $\text{ReAct}_{\text{GPT3.5Turbo}}$     | 23.2     |
> > > | $\text{Plan-RAG}_{\text{GPT3.5Turbo}}$  | 35.4     |
> > >
> > > Additionally, we analyzed the ReAct loops for the above experiment. The histogram is presented below:
> > >
> > > | Loops | Count    | Correct|
> > > | ----- | ------   |---------|
> > > | 0     | 326      | 0   |
> > > | 1     |  94      | 27  |
> > > | 2     | 312      | 165 |
> > > | 3     |  90      | 28  |
> > > | 4     |  76      | 12  |
> > > | 5     | 102      | 0   |
> > >
> > > These results highlight the computational benefits of Plan-RAG over ReAct (see Table 2 of the manuscript).

---

> > > > ### Author Response · Authors · 2024-11-28
> > > >
> > > > We thank the reviewer for their thoughtful engagement. We would appreciate hearing their perspective on the comparison experiment between ReAct and Plan-RAG outlined above.

---

### Official Review · Reviewer_psoq · 2024-11-01

**Soundness:** 1
**Presentation:** 2
**Contribution:** 2
**Rating:** 3
**Confidence:** 4

**Summary:**

This paper proposes a plan-based RAG model, called Plan-RAG, which decomposes a query into multiple subqueries and constructs a directed acyclic graph (DAG) based on the subqueries' hierarchy. The RAG system, using multiple LLM experts, traverses the graph to respond to the original query. Empirical experiments demonstrate that the proposed approach achieves competitive performance compared to other state-of-the-art models.

**Strengths:**

- The paper proposes a systematic approach to decomposing queries to address problems requiring multi-hop reasoning, enhancing the system's trustworthiness and interpretability.

- The proposed approach employs multiple LLM experts to make the system more efficient and effective.

**Weaknesses:**

- Although the system applies DAG, the proposed approach doesn’t seem to offer a novel idea compared to other query decomposition methods (such as RQ-RAG and RA-ISF); it feels more like an incremental engineering effort.

- The experiments are not conducted thoroughly:
  - The authors use up-to-date LLMs, such as GPT-4 and LLaMA 3, to implement Plan-RAG, while using LLaMA 2 for other models (Self-RAG, RQ-RAG), which significantly hinders a fair comparison. Therefore, it would be beneficial to evaluate the comparative approaches using the same base models as those employed by Plan-RAG.
  - Since there are no comparative evaluations of Attribution, it is difficult to assess the performance of the proposed Plan-RAG. It would be desirable to compare Plan-RAG’s performance with other approaches in terms of Attribution.
  - Although the Critic Expert reduces the number of retrievals, the resulting performance is not promising. Additional analysis may be needed to assess how effectively the reduced retrievals provide context.


Typos and presentation errors:
- Line 238: q —> q ̃
- Line 279: retrieval(s) might be confusing.
- Line 314: There should be a space before ‘More’.
- Table 6 is only referenced in the appendix and not in the main pages.

**Questions:**

What is the reason the authors use different base LLMs to compare the proposed Plan-RAG with other state-of-the-art models, such as Self-RAG and RQ-RAG?

---

> ### Author Response · Authors · 2024-11-19
>
> We thank the reviewer for their comments. We address the concerns below.
>
> **(W1) Novelty compared to query-decomposition:** Plan-RAG introduces key differences from query-decomposition methods like RQ-RAG and RA-ISF:
> * *Structured Reasoning Plan:* Plan-RAG generates a structured reasoning plan represented as a DAG, establishing interrelationships between subqueries to support organized and efficient retrieval. This structured approach directs a constrained and relevant flow of information to each subquery, preventing context-window overflow.
> * *Enhanced Attribution due to atomic queries:* Unlike other query-decomposition methods that decompose queries into subqueries of similar complexity, Plan-RAG breaks down queries into interrelated, subatomic queries. Each subatomic query is precise enough to find its answer within a single document, facilitating attribution.
> * *Parallelizable subqueries:* The DAG structure enables subqueries to be processed in parallel, significantly improving response time. In contrast, traditional methods handle subqueries sequentially, which is inherently slower.
>
> **(W2) Up-to-date LLMs as experts:** Plan-RAG's plug-and-play architecture allows seamless use of the latest LLMs, like Llama3-8b-Instruct, as experts. In contrast, methods like Self-RAG and RA-ISF rely on fine-tuned Llama-2 models distilled from GPT models. This distillation process is costly, computationally intensive, and requires specialized expertise. Additionally, as LLMs evolve, these methods cannot directly adapt, while Plan-RAG offers more flexibility to incorporate advancements in LLMs.
>
> **(W3) Attribution evaluation:** Plan-RAG achieves built-in attribution by design. Each subquery in the reasoning DAG is subatomic and, when retrieval is triggered, retrieves only a single document, creating a direct one-to-one mapping between each subquery and its corresponding document. This design enables clear attribution. Other methods, however, retrieve a top-k set of documents (*e.g.*, k=5 or 10), over which the LLM reasons to generate a response, leading to ambiguous attribution. Reducing k to 1 for these methods results in a substantial accuracy drop, underscoring Plan-RAG’s advantage in balancing attribution with performance.
>
> **(W4) Critic expert performance:** In Section 4.2 (Table 6 in the manuscript), we conducted an ablation study to evaluate the impact of the Critic Expert. The results indicate that removing the Critic Expert and always retrieving results in a minor accuracy difference of only 0.53%, while significantly increasing the number of retrievals by 600 (from 2530 to 3163). This suggests that the Critic Expert effectively reduces the number of retrievals with minimal impact on accuracy. However, we acknowledge that further analysis is needed to assess how well the reduced retrievals provide sufficient context for generating high-quality responses. Future work will focus on a deeper investigation into the context adequacy provided by the Critic Expert's reduced retrievals.

---

> > ### Comment · Reviewer_psoq · 2024-11-23
> >
> > The authors do not appear to have adequately addressed my concern regarding the fair comparison (i.e., the use of GPT-4 and LLaMA 3 versus LLaMA 2), which is crucial to the soundness of the paper. Therefore, I will maintain my original score.

---

> > > ### Author Response · Authors · 2024-11-25
> > >
> > > We would like to state two points in response:
> > >
> > > **Consistency in Base Models:** We acknowledge the importance of using the same base models for a fair comparison. In our experiments, we utilized up-to-date LLMs like GPT-4 and LLaMA3 for Plan-RAG to demonstrate the framework's ability to seamlessly integrate the latest advancements in LLM technology. This choice was made to highlight the flexibility and future-proof nature of Plan-RAG.
> > >
> > > **Challenges with Older Models:** Methods like Self-RAG and RA-ISF rely on fine-tuned LLaMA-2 models, which were distilled from GPT models. This distillation process is not only costly and computationally intensive but also requires specialized expertise. As LLMs evolve, these methods face significant challenges in adapting to newer models without undergoing extensive re-training and fine-tuning.

---

### Official Review · Reviewer_1AyB · 2024-11-04

**Soundness:** 2
**Presentation:** 2
**Contribution:** 3
**Rating:** 5
**Confidence:** 4

**Summary:**

The paper introduces an approach that enhances RAG by shifting from a retrieve-then-reason to a plan-then-retrieve paradigm, which improves efficiency and accuracy in handling complex queries through structured query decomposition. Contributions include:

- A directed acyclic graph (DAG) structure for decomposing complex queries into interrelated sub-queries, enabling information sharing and parallelized generation.
- Integration of "plug-and-play" experts, such as relevance and critic experts, to dynamically assess and refine retrievals without fine-tuning language models, reducing hallucinations.
- Improved attribution by design, as the DAG structure allows clear tracing of generated answers back to specific retrieved documents.

**Strengths:**

* Originality: The paper introduces a new plan-then-retrieve approach, which is a novel method of structuring complex queries by breaking them into smaller, linked parts using a DAG. This allows the model to handle parts of the query in parallel, speeding up response time and reducing errors. The addition of modular "plug-and-play" components, like relevance and critic experts, makes the framework adaptable to different models without retraining.
* Quality: The work is properly evaluated, with tests on multiple QA datasets that show Plan-RAG outperforms other RAG models in accuracy and relevance. Detailed experiments also demonstrate how each component (like the relevance and critic experts) contributes to the overall performance.
* Clarity: The authors explain the motivation behind their approach, and the experimental setup and findings are presented clearly.
* Significance: Plan-RAG could have an impact on how we use retrieval-augmented models, especially in fields where accuracy and traceability are crucial, like healthcare or financial services. The DAG structure could also inspire more structured reasoning approaches in other AI research areas.

**Weaknesses:**

1. The presentation of the overall framework is confusing. In Figure 2, there seems to be 7 agents, but only 4 out of them are explained in Section 3.2. It would be much clearer to present each of the components in parallel and to include an easy example to explain how each component works.

2. Comparison between the proposed method against Self-RAG and RQ-RAG is not fair. “We use the officially released code and associated models for both of these methods.” (L341) It does not make sense to “use the associated models for these methods”. To compare the RAG framework, the backbone models should be consistent. But Plan-RAG leverages Llama3-8b-Instruct, which is itself much stronger than Llama-2-7b and Llama-2-13b, used by the other two RAG baselines. Moreover, Plan-RAG utilizes GPT4o to generate the DAG, which is an essential step for the success of the entire framework.

3. While the modular "plug-and-play" experts add flexibility, they also introduce additional processing steps that require multiple LLM calls and make the overall system slower than single retrieval-then-reason paradigm, especially for real-time applications. The framework’s reliance on multiple experts could make it less suitable for time-sensitive tasks. To address this, the authors could evaluate the system’s runtime in more detail and consider simplifying the expert system or limiting the experts used per task, which would maintain performance without adding significant delays.

4. Plan-RAG’s DAG-based decomposition seems unnecessary for straightforward RAG use cases where queries are single-hop or require less complex reasoning. The paper could be strengthened by identifying specific types of tasks or scenarios where Plan-RAG’s complexity is justified. For broader usability, the authors could explore a simplified, “light” version of the framework that omits some complexity for single-hop or low-complexity queries.

5. The overall effectiveness heavily relies on constructing an accurate DAG that correctly represents the query’s reasoning process. However, creating such a structure automatically, especially for diverse or ambiguous queries, seems itself a challenging task. More experiments or analysis on the reliability and feasibility of automated DAG generation should be included, as would suggestions for how to handle or correct inaccurate DAGs. The authors could consider discussing potential fallback strategies when the DAG construction fails or is incorrect, or they could investigate ways to make the DAG-building process more robust.

**Questions:**

Why does the ablation study only focus on two experts? What about the importance of other experts?

---

> ### Author Response · Authors · 2024-11-19
>
> We thank the reviewer for their comments. We address the concerns below.
>
> **(W1) Figure 2 presentation:** The three additional components in Figure 2 are the retriever, generator, and reasoning planner. These are not included as experts because they serve distinct roles: the retriever is a standard module for fetching relevant documents from the database, the generator is the LLM responsible for answer generation, and the reasoning planner orchestrates task flow between experts, as described in Section 3.1. Conversely, Section 3.2 focuses on experts that handle specialized tasks within the framework. We agree that adding a concrete example will further clarify the interactions between components, and we will incorporate this in the manuscript to provide a clear understanding of the framework.
>
> **(W2) Inconsistent backbone:** Thank you for raising this point. Plan-RAG's plug-and-play architecture allows it to utilize the latest LLMs, like Llama3-8b-Instruct, as experts. On the other hand, methods like Self-RAG rely on fine-tuned Llama-2 models that distill information from GPT models. This distillation process is costly, computationally intensive, and requires specialized expertise. Therefore, simply substituting Plan-RAG's experts with vanilla Llama-2 models would not yield a fair comparison, as Self-RAG and RQ-RAG employ fine-tuned models. Also, as LLMs evolve, these methods can not directly adapt, while Plan-RAG is more flexible. We acknowledge that this aspect could have been more explicitly clarified in the manuscript, and we will update the discussion to clarify these distinctions.
>
> **(W3) Latency of plug-and-play architecture:** Self-RAG, RA-ISF, and RQ-RAG all methods involve multiple calls to the LLM models so the latency of Plan-RAG is comparable (or even better attributed to parallelizable subqueries) with other methods. We discuss the complexity of Plan-RAG in Section 3.2 in the manuscript.
>
> **(W4) DAG complexity for simpler queries:** In the single-hop *PopQA* dataset, the depth of the reasoning DAG is 0 for the majority of queries (see Table 2 in the manuscript), demonstrating Plan-RAG’s ability to adapt to the complexity of each query.
>
> **(W5) Challenges in generating an accurate DAG:** We appreciate the reviewer’s attention to the critical role of accurate DAG construction. The reasoning planner indeed serves as the central component of our framework, which is why we currently utilize GPT-4o for generating the reasoning DAG, as it achieves consistently reliable results. Llama models, without fine-tuning, do not meet this performance threshold; however, once fine-tuned, they could replace GPT-4o in this step.
>
> **(Q) Ablation on only 2 experts:** Plan-RAG comprises four experts: the Dynamic Query Expert, Critic Expert, Relevance Expert, and Aggregator. Our ablation study focuses on the Critic and Relevance Experts, as these two are foundational to the system's performance. The Dynamic Query Expert and Aggregator have more specific, contained roles: the former modifies the query with parent information, while the latter aggregates outputs from various subqueries to form the final answer. Given the limited responsibilities and scope of these experts, we chose not to perform ablations on these components.

---

### Comment · Area_Chair_MDt2 · 2024-11-25

Dear Reviewers,

The rebuttal discussion period is coming to a close and the paper currently has a mix of positive and negative reviewers. The authors have spent a lot of time responding to each concern -- can you take a look at the author responses and let them know any remaining concerns you have?

Best,
AC

---

### Meta-Review · Area_Chair_MDt2 · 2024-12-22

**Metareview:**

This paper introduces Plan-RAG, a plan-then-retrieve framework that uses a reasoning DAG to break down queries, aiming to improve efficiency, attribution, and multi-hop reasoning. The framework relies on modular "plug-and-play" components, delivering strong performance on QA datasets without requiring fine-tuning, and works particularly well for multi-hop reasoning tasks. The reasoning DAG allows for structured query breakdown, parallel processing, and better attribution, while the modular design adds flexibility and scalability.
 However, reviewers concerned about unfair baseline comparisons with advanced LLMs, limited analysis of DAG generation, and insufficient testing of error attribution, and scalability across model sizes. While Plan-RAG shows potential as a useful approach for handling complex queries, addressing these issues with fairer comparisons, better error analysis, and broader testing could greatly improve its impact.

**Additional Comments On Reviewer Discussion:**

Reviewers appreciated Plan-RAG's novel DAG-based framework and improvements in attribution and efficiency but raised concerns about unfair baseline comparisons, limited analysis of DAG generation, scalability, and performance on single-hop tasks. While the rebuttal addressed some issues, reviewers still concerned  about fairness, novelty, and incomplete evaluations, leading to mixed feedback.

---

### Decision · Program_Chairs · 2025-01-22

Reject